# Development of an *in vivo* syngeneic mouse transplant model of invasive intestinal adenocarcinoma driven by endogenous expression of *Pik3ca*[H1047R] and *Apc* loss

**Francesc de las Heras**[1,2], **Camilla B. Mitchell**[1,2], **William K. Murray**[3], **Nicholas J. Clemons**[1,2], **Wayne A. Phillips**[1,2]*

1 Department of Cancer Research, Peter MacCallum Cancer Centre, Melbourne, Victoria, Australia, 2 Sir Peter MacCallum Department of Oncology, The University of Melbourne, Parkville, Victoria, Australia, 3 Department of Pathology, Peter MacCallum Cancer Centre, Melbourne, Victoria, Australia

* wayne.phillips@petermac.org

## Abstract

Preclinical models that replicate patient tumours as closely as possible are crucial for translational cancer research. While *in vitro* cancer models have many advantages in assessing tumour response therapy, *in vivo* systems are essential to enable evaluation of the role of the tumour cell extrinsic factors, such as the tumour microenvironment and host immune system. The requirement for a functional immune system is particularly important given the current focus on immunotherapies. Therefore, we set out to generate an immunocompetent, transplantable model of colorectal cancer suitable for *in vivo* assessment of immune-based therapeutic approaches. Intestinal tumours from a genetically engineered mouse model, driven by expression of a *Pik3ca* mutation and loss of *Apc*, were transplanted into wild type C57BL/6 host mice and subsequently passaged to form a novel syngeneic transplant model of colorectal cancer. Our work confirms the potential to develop a panel of mouse syngeneic grafts, akin to human PDX panels, from different genetically engineered, or carcinogen-induced, mouse models. Such panels would allow the *in vivo* testing of new pharmaceutical and immunotherapeutic treatment approaches across a range of tumours with a variety of genetic driver mutations.

## Introduction

Colorectal cancer is one of the most common cancers worldwide. Those diagnosed with this disease have considerable morbidity and mortality [1,2] with 20% of those diagnosed having metastatic disease and 40% suffering a recurrence following previously treated localized disease [3]. Despite improved treatment options, most patients with metastatic colorectal cancer still die of their disease [3]. Thus, more effective treatments for patients with late-stage colorectal cancer are urgently required.

Most colorectal cancers develop in the epithelium as a consequence of genetic and/or epigenetic alterations that activate the Wnt pathway, largely through the inactivation of the APC

**Data Availability Statement:** All relevant data are within the paper and its Supporting Information files.

**Funding:** F.dlH. was a recipient of an International Postgraduate Research Scholarship from the University of Melbourne.

**Competing interests:** Wayne A. Phillips is a member of the editorial board of PLOS ONE. This does not alter our adherence to PLOS ONE policies on sharing data and materials. The authors have declared that no other competing interests exist.

gene [4]. This initially leads to the formation of benign lesions known as adenomas that, over time, acquire further molecular abnormalities that can transform them into invasive carcinomas and, ultimately, metastatic disease [4,5].

To facilitate translational research, it is essential to have appropriate preclinical models in which to test new therapeutic approaches. Typically, such models involve *in vitro* cell culture systems using human or mouse cancer cell lines [6–8]. More recently, three-dimensional organoid culture systems using primary patient-derived cells have become available [9,10]. While such *in vitro* cancer models have many advantages, *in vivo* systems are essential to evaluate the role of the tumour cell extrinsic factors, such as the tumour microenvironment and host immune system, in the tumour's response to therapy.

To address this, many murine models of colorectal cancer have been reported, each with their own advantages and disadvantages [6–8]. Carcinogen-induced and genetically engineered models are perhaps the most physiologically relevant models, developing *in situ* and reproducing the early stages of oncogenesis. However, the internal location of the tumours and the long latency period makes monitoring tumour development and assessing response to therapy difficult. Also, these tumours often quickly impact the health of the mouse, requiring euthanasia for ethical reasons, before the full extent of progression to adenocarcinoma or the full impact of treatment can be adequately achieved.

The subcutaneous transplantation of tumour cell lines or patient tumour tissue into mice allows for simple and regular monitoring of growth but the use of tissue samples and cell lines of human origin requires immunocompromised hosts meaning the impact of the immune system cannot be fully evaluated.

The requirement for a functional immune system is particularly important given that many new approaches to cancer treatment are focussing on the potential use of immunotherapies. Therefore, we set out to generate an immunocompetent model suitable for *in vivo* assessment of immune-based approaches for the treatment of colorectal cancer. We have previously demonstrated that a combination of *Pik3ca*$^{H1047R}$ mutation and *Apc* loss is sufficient drive the formation of intestinal tumours in a mouse model [11]. Here we have generated mouse intestinal tumours on a pure C57BL/6 background by expressing *Pik3ca*$^{H1047R}$ mutation and *Apc* loss in *Lgr5*-positive intestinal stem cells. These tumours were used to generate a subcutaneous syngeneic graft model of colorectal cancer as an immunocompetent model suitable for assessing immunotherapies.

## Materials and methods

### Animal husbandry

Compound mice harbouring a latent Cre-inducible knock-in of the *Pik3ca*$^{H1047R}$ mutation (*Pik3ca*$^{Lat-H1047R}$) [12], homozygous *Cre*-inducible *Apc* loss-of-function alleles (*Apc*$^{580S/580S}$) [13], and a tamoxifen-activatable *Lgr5*-driven Cre recombinase (*Lgr5-CreERT2*) allele [14] were generated on a pure C57BL/6 background. To induce Cre-recombination, tamoxifen was administered via a single intraperitoneal injection at a dose of 2 mg/mouse, at 6–8 weeks of age resulting in mice expressing a heterozygous *Pik3ca*$^{H1047R}$ mutation and homozygous *Apc* loss of function (now *Apc*$^{580D/580D}$ when alleles were deleted) in the gastrointestinal tract. All mice were routinely monitored post tamoxifen administration and humanely euthanised by cervical dislocation or carbon dioxide asphyxiation for tumour collection when there was a >20% loss in body weight or any signs of distress (bloated, pale feet, ruffled fur, lack of movement, etc.). Wildtype C57BL/6 mice were purchased from the Walter and Eliza Hall Institute of Medical Research (Melbourne, Australia). All mice were fed a standard chow diet (irradiated Barastoc mouse cubes; Ridley AgriProducts, Melbourne, Australia).

All animal experiments were approved by the *Peter MacCallum Cancer Centre Animal Experimental Ethics Committee* (Project Number E584) and conducted in accordance with the *National Health and Medical Research Council Australian Code of Practice for the Care and Use of Animals for Scientific Purposes.*

## Tumour transplantation

Primary tumours expressing a heterozygous *Pik3ca*$^{H1047R}$ mutation together with a homozygous loss of *Apc* (*Apc*$^{580D/580D}$) were derived from tamoxifen-treated *Lgr5-CreERT2:Pik3ca-*$^{Lat-H1047R}$:*Apc*$^{580S/580S}$ mice. The gastrointestinal tract was removed from these mice post-euthanasia. The small intestine and colon were flushed clean with phosphate-buffered saline (PBS) and any tumours present were removed and transplanted either intramuscularly or subcutaneously into wild type C57BL/6 mice.

For intramuscular transplantation, small pieces of tumour were transplanted intramuscularly as previously described [15]. Mice were anesthetized via an intraperitoneal injection of 100 µl of anaesthetic solution (ketamine 10 mg/ml and xylazine 2 mg/ml) per 10 g of body weight. The dorsum of the mouse was shaved and prepared with a 2% (v/v) chlorhexidine gluconate/70% (v/v) isopropyl alcohol solution. A 15 mm incision was made along the dorsal midline of the mouse and a pocket was created in the dorsal musculature using a combination of sharp and blunt dissection until it was large enough to accommodate the tumour piece. The tumour piece coated in Matrigel (Corning Inc., Corning, NY) was then placed in the pocket. The skin was then closed using wound clips which were removed after 10 days [15]. Two separate transplantation sites were used per mouse, one each side of the spinal column [15].

For subcutaneous transplantation, mice were anaesthetized, the dorsum of the mouse prepared as above, and a 15mm midline incision made along the dorsal upper line of the mouse. A small section of skin was mechanically detached with the tip of dissecting tenotomy scissors to accommodate the tumour piece. The tumour piece coated in Matrigel was then placed under the skin. The skin was then closed using wound clips which were removed after 10 days. Two separate transplantation sites were used per mouse, one on each side of the spinal column.

Mice were monitored daily for any changes in their health, and tumours were measured twice a week with callipers once they became palpable. Tumour volume was calculated by the formula (length x width$^2$)/2 and mice were euthanised when the total tumour volume reached ethical limit (1400 mm$^3$), or they became distressed or moribund (whichever came first).

## Tumour collection

Following euthanasia by cervical dislocation or $CO_2$ asphyxia, tumours were excised and rinsed with PBS containing 100 U/ml Penicillin, 100 µg/ml Streptomycin and 100 µg/ml Gentamicin. Half of the tumour was fixed in 10% neutral-buffered formalin (NBF) overnight at 4°C, dehydrated through a series of ethanol dilutions, and embedded in paraffin for histology and immunohistochemistry (IHC). The other half of the tumour was dissected into small pieces (10 mm$^3$), half of which were placed in 1.5 ml Eppendorf tubes (Hamburg, Germany), snap frozen in liquid nitrogen and stored at -80°C for protein and RNA extraction. The remaining pieces were placed in either ice cold Matrigel for a fresh transplantation, or room temperature freezing media (10% DMSO (Millipore, Burlington, MA) and 90% foetal bovine serum (FBS, Thermo Fisher Scientific, Waltham, MA) for cryopreservation. Vials containing samples for cryopreservation were slowly cooled to -80°C in a Mr Frosty™ cryogenic container (Thermo Fisher) with isopropanol and then stored in liquid nitrogen.

## Immunohistochemistry

Paraffin embedded tissues and tumours were sectioned at 4 μm thickness. Sections were stained with haematoxylin and eosin (H&E) for histological analysis by an anatomical pathologist. Sections were also stained with the following antibodies: phospho-S6 ribosomal protein (Ser235/236) (1:200; Cell Signalling Technologies, Danvers, MA; #2211), β-catenin (1:1000; BD Biosciences, Franklin Lakes, NJ; #610153), Ki67 (1:100; Abcam, Cambridge, UK; #Ab16667), lysozyme (1:600; Abcam; #Ab108508), Muc2 (1:200; Santa Cruz Biotechnology, Santa Cruz, CA; #Sc-15334). Signals were detected using DAKO secondary antibodies, and the DAB+ chromogen kit (Vector Laboratories, Burlingame, CA). Slides were scanned using an Olympus VS120 slide scanner with a 20x objective.

## RNA extraction and RT-PCR analysis

Allele-specific RT-PCR, targeting the mutated codon together with other silent base changes engineered into the mutant exon [12], was used to confirm knock-in of the *Pik3ca* mutation. RNA was extracted from snap frozen mouse tumour samples using a Nucleospin RNA kit (Macherey-Nagel, Allentown, PA) and following manufacturer's protocols. RNA was transcribed to cDNA using the Transcriptor First Strand cDNA Synthesis Kit (Roche Diagnostics Australia, North Ryde, Australia). cDNA (2 μl) was mixed with 1 μl of 10x Qiagen Buffer (Qiagen, Hilden, Germany), 0.5 μl of 2.5 mM dNTPs (Sigma-Aldrich, Burlington, MA), 0.45 μl of 10 μM *Pik3ca* common forward primer (5'-CAAGAGTACACCAAGACCAGAGAGTT-3') (GeneWorks, Adelaide, Australia) and 0.45 μl of either 10 μM *Pik3ca* wild type-specific reverse (5'-TGTCGTCCATCCACCATGATGT-3') or 10 μM *Pik3ca* mutant-specific reverse (5'-TGTCGTCCACCCTCCGTGCCTA-3') primer (GeneWorks), 0.05 μl of 5 U/μl Qiagen HotStarTaq® DNA Polymerase, and made up to 10 μl with milliQ $H_2O$, and amplified at 95 ˚C for 10 min, followed by 95 ˚C for 30 sec, 57 ˚C for 30 sec, 72 ˚C for 40 sec, for 35 cycles, and then for 72 ˚C for 5 min. The amplified products were run on 1.5% agarose gel (in 1x tris-acetate-EDTA (TAE) buffer) at 150 V for 45 min and stained with Midori Green DNA stain (Nippon Genetics, Düren, Germany). Gels were imaged on a Chemidoc system, using Image Lab software.

## Drug treatment

Tumours were transplanted into wild type C57BL/6 mice (1 tumour per mouse). Once tumours reached 50–200 mm$^3$ in volume, mice were randomly assigned to be treated with BYL719 (MedChemExpress, Monmouth Junction, NJ; 50 mg/kg in 0.5% methylcellulose (Sigma-Aldrich)) or vehicle control (0.5% methylcellulose) by daily oral gavage. Mice were closely monitored, and tumour size was measured regularly. Mice were euthanised when the tumours reached ethical limit ($\geq$1400 mm$^3$) or if the mice were visibly unwell or moribund.

# Results

## Generation of a syngeneic model of intestinal adenocarcinoma

*Lgr5-CreERT2:Pik3ca$^{Lat-H1047R}$:Apc$^{580S/580S}$* mice were treated with a single dose of tamoxifen to induce expression of the *Pik3ca*$^{H1047R}$ and *Apc*$^{580D/580D}$ alleles in the gastrointestinal epithelium. At first sign of discomfort or distress (mean 6 weeks post tamoxifen), the mice were euthanised and the gastrointestinal tract examined for tumours. Representative tumours (termed "parental") were selected for transplant into wild type C57BL/6 host mice.

Tumour pieces were transplanted intramuscularly, rather than subcutaneously, as we have previously shown that intramuscular transplantation improves initial engraftment of tumour grafts [15], presumably because of the well-vascularized transplant bed provided by the muscle

tissue. Once established, tumours were sequentially passaged through two further generations (F1 & F2) intramuscularly, followed by transplantation of the third generation (F3) subcutaneously, to further facilitate tumour size monitoring.

## Tumours become progressively more dysplastic and aggressive over transplantation generations

In order to classify and grade the tumours from each generation, H&E histology sections from wildtype small intestine, parental tumours, F1, F2 and F3 tumours, were examined by an anatomical pathologist (WKM) (Fig 1). Images from wildtype mice show typical normal small intestine with muscularis propria, submucosa, and the mucosa adjacent to the lumen (Fig 1A). *Lgr5-Cre*: *Pik3ca*$^{H1047R}$:*Apc*$^{580D/580D}$ parental tumours were classified as adenomas displaying high-grade dysplasia throughout the polyp mass. These adenomas present with an aggressive phenotype, resembling adenocarcinomas, but without tumour invasion into the underlying submucosa and muscle layer (Fig 1B). F1 transplanted tumours are also classified as adenomas displaying high-

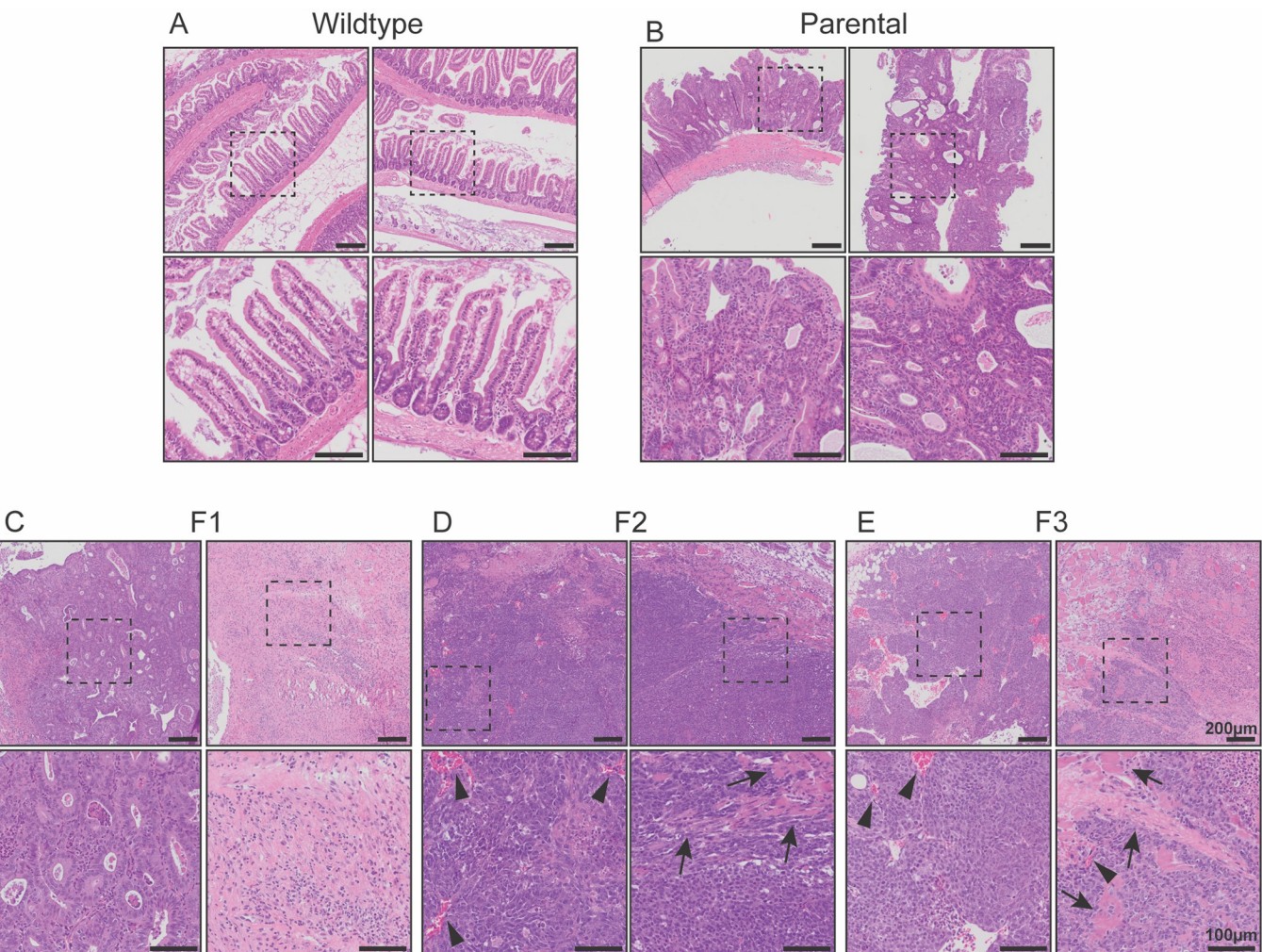

**Fig 1. Histopathology of parental tumour and filial syngeneic grafts.** Representative H&E images of **A.** wild type small intestine, **B.** *Pik3ca*$^{H1047R}$:*Apc*$^{580D/580D}$ parental tumours, **C.** F1, **D.** F2, and **E.** F3 syngeneic grafts. Two example areas are shown for each graft. Areas outlined in dashed lines are magnified in boxes below. Arrow heads highlight blood vessel invasion, arrows highlight muscle invasion. Scale bars represent 200 μm (original images) and 100 μm (magnified images).

grade dysplasia, with no clear signs of invasion into the surrounding muscle. Of note, many F1 tumours consisted of liquid-filled cystic tissue displaying fibrosis and inflammation (Fig 1C, lower right panel), as well as sections of solid, adenomatous tissue (Fig 1C, lower left panel). When transplanting from F1 to F2, the adenomatous and not cystic tissue was selected to increase chances of successful tumour growth. The majority of F2 tumours were solid, with only one developing a cyst. These solid tumours were classified as poorly differentiated aggressive adenocarcinomas, with both muscle (arrows) and blood vessel (arrow heads) invasion (Fig 1D). To extend the model, the most aggressive F2 tumours were selected to be transplanted into F3 mice. All F3 tumours were subcutaneously transplanted, rather than intramuscularly, due to the increased aggressiveness of the tumours, and success rate of tumour transplantation and growth. F3 tumours were classified as very poorly differentiated, aggressive adenocarcinomas, displaying high-grade dysplasia, and both muscle (arrows) and blood vessel (arrow heads) invasion. Variable levels of necrosis and fibrosis are also seen throughout these tumours (Fig 1E).

We compared transplantation efficiency using fresh or cryopreserved tissue (S1A Fig). When transplanting F1 tumours, 14 out of 20 (70%) were successful with fresh tissue and with cryopreserved tissue 3 out of 5 (60%) were successful. Finally, in F3, 13 transplantations were performed with fresh tissue and 6 with cryopreserved tissue. Moreover, there was no significant difference in growth rate, as defined by the average days to ethical limit, between transplants from fresh versus cryopreserved tissue (S1B Fig).

## Confirmation of $Pik3ca^{H1047R}$ mutation knock-in

Since these tumours are all derived from an original $Lgr5$-$Cre$:$Pik3ca^{H1047R}$:$Apc^{580D/580D}$ tumour, where a H1047R mutation has been endogenously knocked-in to the $Pik3ca$ allele, we used an allele-specific RT-PCR approach to confirm the presence of the knocked-in mutation across transplantation generations (Fig 2A). Small intestine tissue from a non-tamoxifen treated (negative control) $Lgr5$-$CreERT2$:$Pik3ca^{Lat\text{-}H1047R}$ mouse shows a 715 bp product, and no 250 bp product, with mutant-specific primers (Fig 2B). In contrast, $Pik3ca^{H1047R}$ knock-in was confirmed to be present across F1, F2 and F3 tumours, as shown by the 255 bp product when mutant primers were used, indicating that once the mutation is knocked-in, it remains expressed across generations of tumour transplantation (Fig 2B).

To confirm that the $Pik3ca^{H1047R}$ mutation confers PI3K pathway activity, we performed IHC staining on the downstream PI3K target, pS6, which is routinely used as a marker of PI3K activity. While pS6 is restricted to the tips of villi in wild type small intestine, the expression levels of pS6 increases, and is spread all throughout the tumour tissue in both parental adenomas and F3 adenocarcinomas (Fig 3A).

APC deletion was confirmed indirectly through IHC, as demonstrated by a significant accumulation of cytoplasmic β-catenin. Wild type tissue shows a predominately membranous staining of β-catenin throughout the small intestine epithelium, whereas parental adenomas show an increase in β-catenin expression levels, as well as a change in localisation from the membrane to the cytoplasm. F3 tumours exhibit more intense staining, with some instances of nuclear staining (Fig 3B). Increased tumour aggressiveness through in vivo passaging corresponded with increased proliferation, identified by Ki67-positivity (Fig 3C). Ki67 proliferative cells are confined to the base of the crypts of wild type intestinal epithelium but are present throughout the parental and F3 tumour tissues (Fig 3C).

Lysozyme staining was performed to label the Paneth cells which are normally situated in the crypt of the villi next to the stem cells, as seen in the wild type small intestine (Fig 3D). Additionally, Mucin2 staining was used to label the goblet cells, normally situated in the villi and crypts of the normal intestine (Fig 3E). Interestingly, with respect to both markers, the

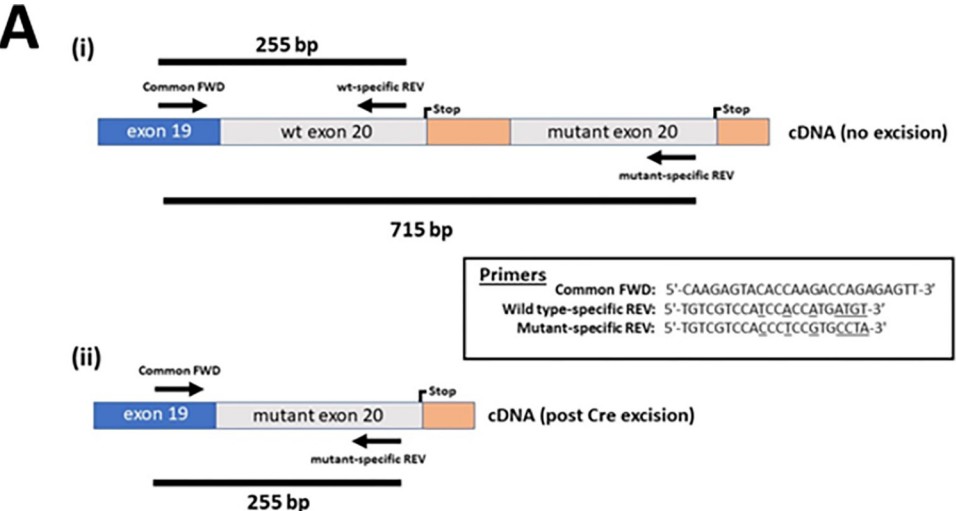

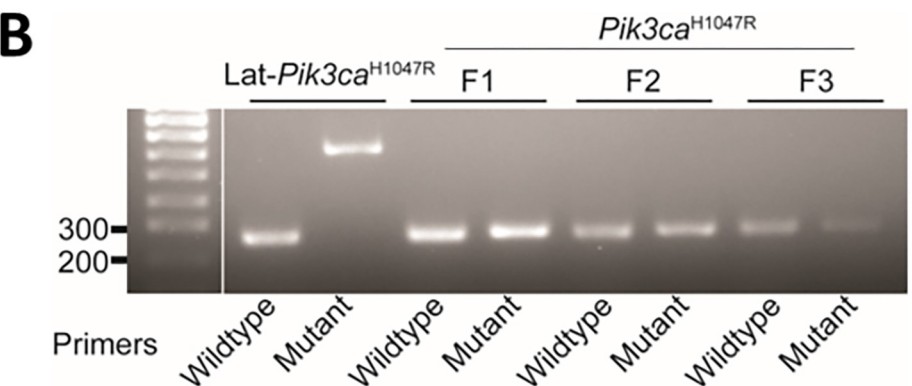

**Fig 2. Confirmation of *Pik3ca*[H1047R] mutation knock-in.** Expression of the mutant *Pik3ca*[H1047R] allele was detected using an allele-specific RT-PCR-based strategy targeting the mutated codon and silent base changes engineered into the mutant codon [12]. (a) Schematic diagram of the cDNA structure before and after Cre-mediated excision showing the relative positions of exon 19 (blue box), wild type (wt) and mutant exon 20 (grey boxes), and stop codons (Stop). The locations of the respective primer binding sites are indicated by arrows and the PCR fragment sizes indicated by the black bars. (b) cDNA from un-excised control mouse intestinal tissue (Lat-*Pik3ca*[H1047R]) and mouse tumours (F1, F2 and F3) were amplified using allele-specific primers for the wild type and mutant alleles (as indicated) and the amplified product run on 2% agarose gel and stained with ethidium bromide.

parental adenoma tissue showed an overall heterogeneous pattern (Fig 3D and 3E). In contrast, F3 tumour tissue was negative for these markers, consistent with a lack of cell differentiation, as suggested by the pathological observations. The gradual loss of mucin producing goblet cells over serial transplantation might explain the loss of the fluid-filled cysts observed in the initial (F1) transplantations over later generations.

## Tumour growth is repressed by treatment with a PI3Kα-specific inhibitor, BYL719

To demonstrate the utility of our model to test potential therapies for colorectal cancer, we selected the PI3Kα-selective inhibitor, BYL719, to treat subcutaneously transplanted *Pik3ca*-mutant adenocarcinomas.

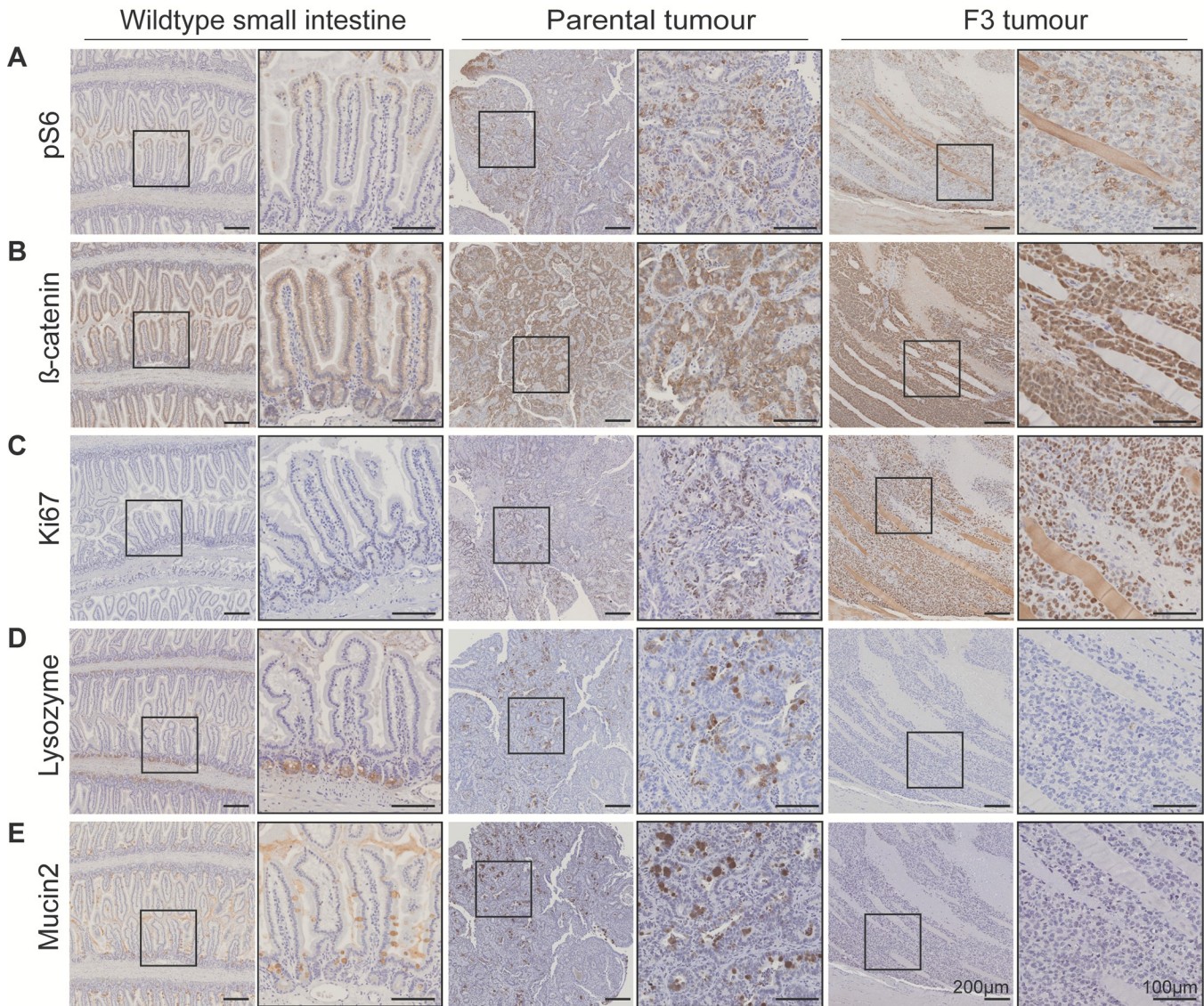

**Fig 3. Representative immunohistochemistry images of wildtype small intestine epithelium, *Pik3ca*[H1047R]:*Apc*[580D/580D] parental tumours, and F3 tumours.** Staining for **A.** pS6 (marker of PI3K signalling), **B.** β-catenin (marker of active Wnt signalling), **C.** Ki67 (marker of cell proliferation), **D.** Lysozyme (marker of Paneth cells), and **E.** Mucin2 (marker of goblet cells). Areas outlined in dashed lines are magnified in adjacent boxes. Scale bars represent 200μm (original images) and 100μm (magnified images).

We transplanted cryopreserved tissue from an F3 tumour into female C57BL/6 mice. Six tumour-bearing mice were randomly assigned into two groups and administered either 50 mg/kg BYL719 or 0.5% methylcellulose as vehicle control, via daily oral gavage beginning 20 days post-transplantation. All control mice reached ethical limit by 74 days (Fig 4). In contrast, treatment with BYL719 successfully slowed the rate of growth of tumours, compared to the vehicle control, with none of the BYL719 treated mice reaching ethical limit by 100 days (Fig 4).

## Discussion

In this manuscript we describe the establishment of a novel immunocompetent syngeneic transplant mouse model of colorectal cancer. Gastrointestinal tumours from a genetically engineered

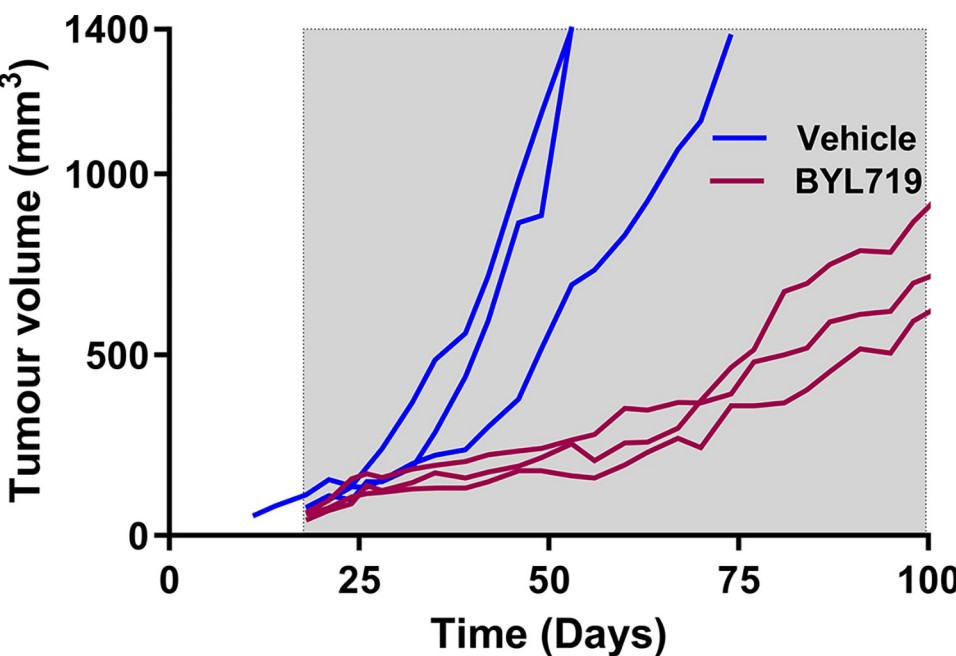

**Fig 4. Tumour growth is repressed by treatment with a PI3Kα-specific inhibitor, BYL719.** F3 tumours were transplanted into wild type C57BL/6 mice. Mice were treated daily with vehicle (0.5% methylcellulose) or BYL719 (50 mg/kg in 0.5% methylcellulose) by oral gavage. Line graphs show tumour volumes (mm³) over time for vehicle control (blue) and BYL719-treated (magenta) tumours. Grey background indicates treatment period.

mouse model, driven by expression of a *Pik3ca* mutation and loss of *Apc*, were transplanted into wild type C57BL/6 host mice. Initial transplantation was intramuscular, to facilitate establishment of the syngeneic graft, but subsequent passaging allowed adaption to a subcutaneous transplantation site and high transplantation efficiency. The utility of this model for assessing the *in vivo* therapeutic activity of a pharmacological agent was demonstrated using the PI3Kα inhibitor, BYL719, which was shown to slow tumour growth compared to the untreated controls.

Although the initial parental tumour was characterised as an adenoma, upon serial passaging there was clear progression to a more aggressive adenocarcinoma phenotype, as characterised by loss of differentiation and muscle and blood vessel invasion. This progression in phenotype could be due to the acquisition of an additional mutation(s) during passaging or, perhaps, the outgrowth of a more aggressive clone of adenocarcinoma cells that was already present in the parental tumour.

Our model overcomes several drawbacks that hamper the utility of many murine models of colorectal cancer [7,8,16–18]. The subcutaneous transplantation approach facilitates monitoring and measuring of the tumour growth that is difficult with internal (*in situ*) gastrointestinal tumour models. Furthermore, the subcutaneous site allows for the growth of larger tumours, and thus longer treatment times, before the health and comfort of the host mouse is impacted, and ethical endpoints are reached. In addition, the transplantation of tumour pieces allows for the maintenance of tumour heterogeneity and the tumour microenvironment, including tumour infiltrating immune cells, that are lost when using cell line models. Moreover, the syngeneic nature of our model allows for the use of an immunocompetent host, an important advantage that is not possible with human patient-derived xenograft models.

For this reason, we believe our model will be a valuable addition to the current range of murine models of colorectal cancer. While we have used a *Pik3ca* mutation/Apc loss driven mouse model to derive our transplantable syngeneic graft lines, we foresee using this approach

to generate a panel of syngeneic models with different mutational drivers. Such a panel would allow the use of genomic and proteomic approaches to explore the mechanism(s) underlying response and/or resistance to specific therapies, and to identify potential predictive biomarkers, akin to previously reported studies on human patient-derived xenograft [19,20] and organoid culture [21,22] models. Importantly, and in contrast to the human models, our host has a fully functional immune system, allowing not only the *in vivo* evaluation of immune-targeted therapies but also allowing the contribution of specific immune cell populations to the response to different therapies to be appropriately assessed. The *in vivo* nature of our model also allows the role of other components of the tumour microenvironment to be studied and the testing potential therapies that target tumour-microenvironment interactions.

Of course, no model system is perfect, and our model also has limitations. Primarily, the tumours are not human in origin. Furthermore, the tumours are not growing in the physiological site but rather subcutaneously, and the subcutaneous environment is vastly different from that of the intestinal epithelium. Also, while using syngeneic tumour grafts maintains a degree of intra-tumoural heterogeneity, the tumours themselves have restricted inter-tumour variability, particularly when derived from mice with specific genetically-engineered driver events.

While we have used a *Pik3ca* mutation/*Apc* loss driven mouse model to derive our transplantable syngeneic graft lines, our work confirms the ability to transplant murine tumours into immunocompetent mouse hosts and establishes the potential to use a range of different genetically-engineered, or carcinogen-induced, mouse models to generate panels of mouse syngeneic grafts, akin to human PDX panels. Such panels would allow the *in vivo* testing of new pharmaceutical and immunotherapeutic treatment approaches across a range of tumours with a variety of genetic driver mutations.

## Supporting information

**S1 Fig. Transplantation efficiency of fresh and cryopreserved tumour tissue. A.** Stacked bar graphs display percentage of transplanted fresh and cryopreserved (Cryo) tumour tissues that generated new tumour growth. Magenta, blue, and orange shading represents the proportion of F1, F2 and F3 tumours, respectively, that grew upon transplantation. White represents the proportion of tumours that did not grow. Total numbers transplanted (n) are shown at the top of each bar. F1 and F2 tumours were transplanted intramuscularly whereas F3 tumours were transplanted subcutaneously. **B.** The number of days taken for tumours to reach ethical limit across each generation of transplantations, comparing fresh and cryopreserved tumour tissue. Magenta = F1, Blue = F2, Orange = F3. Open circles represent fresh tissue, filled circles represent cryopreserved tissue. Boxes represent the mean. Error bars represent SEM.
(TIF)

**S1 Raw images.**
(PDF)

## Acknowledgments

The authors thank the staff of the Animal facility and the Centre for Advanced Histology and Microscopy of the Peter MacCallum Cancer Centre for their valuable advice and/or technical assistance.

## Author Contributions

**Conceptualization:** Francesc de las Heras, Camilla B. Mitchell, Nicholas J. Clemons, Wayne A. Phillips.

**Data curation:** Francesc de las Heras, Camilla B. Mitchell, William K. Murray.

**Formal analysis:** Francesc de las Heras, Camilla B. Mitchell, William K. Murray, Nicholas J. Clemons, Wayne A. Phillips.

**Funding acquisition:** Wayne A. Phillips.

**Investigation:** Francesc de las Heras, Camilla B. Mitchell, William K. Murray, Nicholas J. Clemons.

**Methodology:** Francesc de las Heras, Camilla B. Mitchell, Nicholas J. Clemons, Wayne A. Phillips.

**Project administration:** Wayne A. Phillips.

**Resources:** Nicholas J. Clemons, Wayne A. Phillips.

**Supervision:** Camilla B. Mitchell, Nicholas J. Clemons, Wayne A. Phillips.

**Visualization:** Francesc de las Heras, Camilla B. Mitchell, Wayne A. Phillips.

**Writing – original draft:** Francesc de las Heras.

**Writing – review & editing:** Camilla B. Mitchell, William K. Murray, Nicholas J. Clemons, Wayne A. Phillips.

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
