## [Decision Letter · Decision Letter 0]

28 May 2024

PONE-D-24-19048Development of an in vivo syngeneic mouse transplant model of invasive intestinal adenocarcinoma driven by endogenous expression of Pik3caH1047R and Apc lossPLOS ONE

Dear Dr. Phillips,

Thank you for submitting your manuscript to PLOS ONE. After careful consideration, we feel that it has merit but does not fully meet PLOS ONE’s publication criteria as it currently stands. Therefore, we invite you to submit a revised version of the manuscript that addresses the points raised during the review process.

This is an interesting manuscript well performed and that opens insights on the possible use in cancer therapies. The Authors must highlight this aspect and add some info about human more recent studies on organoids and more recent use in therapies as specified to them by a reviewer and by myself in our reports.

We look forward to receiving your revised manuscript.

Kind regards,

Gianpaolo Papaccio, M.D., Ph.D.

Academic Editor

PLOS ONE

Journal Requirements:

https://link.springer.com/article/10.1007/s00384-018-3202-8?

https://www.pure.ed.ac.uk/ws/files/20002553/Mouse_models_of_colorectal_cancer_as_preclinical_models.pdf?

In your revision ensure you cite all your sources (including your own works), and quote or rephrase any duplicated text outside the methods section. Further consideration is dependent on these concerns being addressed.

"I have read the journal's policy and the authors of this manuscript have the following competing interests: 

Wayne A. Phillips is a member of the editorial board of PLOS ONE. The authors have declared that no other competing interests exist."

**Additional Editor Comments:**

In this study, the Authors obtained an immunocompetent, transplantable model of colorectal cancer, that can be suitable for in vivo studies of immune-based therapies.

Albeit its interest, it must then evaluate the relationships with TME.

Taking into consideration that it is a murine model and not a human one, the Authors must add and focus in the human cancer field in order to give more interest to this study, adding several info on organoids and human colorectal cancer as well as proteo trascriptomics and TME ( J Exp Clin Cancer Res. 2023 Jan 6;42(1):8. doi: 10.1186/s13046-022-02591-z; doi: 10.3390/cancers12123611; ESMO Open Volume 5, Issue 523 September 2020 Article number e000847;International Journal of Molecular Sciences Volume 24, Issue 23December 2023 Article number 17101 etc).

Reviewers' comments:

Reviewer's Responses to Questions

**Comments to the Author**

1. Is the manuscript technically sound, and do the data support the conclusions?

Reviewer #1: Yes

2. Has the statistical analysis been performed appropriately and rigorously? 

Reviewer #1: Yes

3. Have the authors made all data underlying the findings in their manuscript fully available?

Reviewer #1: Yes

4. Is the manuscript presented in an intelligible fashion and written in standard English?

Reviewer #1: Yes

5. Review Comments to the Author

Reviewer #1: In this manuscript, the Authors generated an immunocompetent, transplantable model of colorectal cancer suitable for in vivo assessment of immune-based therapeutic approaches. The manuscript is interesting, and it is a starting point to develop a model by which it is possible to evaluate relationship among different cell types homing in the tumour microenvironment. Albeit this, the authors must better focus on how they want to exploit this model in human cancer field. Moreover, they must add information regarding the possibility to use other models as organoids for drug screening (citing the studies as J Exp Clin Cancer Res. 2023 Jan 6;42(1):8. doi: 10.1186/s13046-022-02591-z; doi: 10.3390/cancers12123611.) and tumour microenvironment evaluation.

6. PLOS authors have the option to publish the peer review history of their article (what does this mean?). If published, this will include your full peer review and any attached files.

Reviewer #1: No

---

## [Author Response · Author response to Decision Letter 0]

15 Jul 2024

Response to reviewer and editor comments

We thank the reviewer and editor for their positive comments and suggestions.

As suggested by the reviewer, we have added some comments to the discussion to better highlight the relevance to human cancer and how we feel our model could be exploited in future studies (lines 315-332 in tracked version).

Regarding evaluating the relationships with TME, we agree that this is a particularly interesting question which our model will be very useful for addressing, and we have now added some comments to this effect in the discussion (lines 326-332 in tracked version). However, we do feel that such studies are beyond the scope of our current manuscript which simply aims to report our model. 

Indeed, we would like to stress this manuscript is intended to simply report the development of a novel syngeneic mouse model of a PIK3CA-driven colorectal cancer. We have not sought to address any specific biological or clinical question in this manuscript. Nevertheless, we believe this model will be a valuable adjunct to current models and we feel it important to report this model to the wider research community in a timely fashion.

We note that the reviewer and editor request that we cite several specific papers. We have read these articles with interest. 

Papaccio et al. (2023) J Exp Clin Cancer Res. 42(1):8. This paper describes the generation of a large number of patient-derived organoid models that they use for a proteotranscriptomic study to explore the mechanisms underlying drug response. Although they use human organoids, and an in vitro cell culture system, we agree that it is an excellent example of how our model could be exploited and have now included reference to this paper in our discussion (reference 22, line 326 of tracked change version). 

Papaccio et al. (2020) Cancers. 12: 3611. This is a review article that discusses two potentially practice-changing clinical trials and how they may affect our understanding of treating locally advanced rectal cancers. An interesting and informative article but unclear how it specifically relates to our current manuscript. 

Tarazona et al. (2020) ESMO Open. 5: e000847. A clinical study of ctDNA and CDX2 expression as markers of recurrence in patients with localised colon cancer but of unclear relevance to our manuscript. 

Papavassiliou et al. (2023) Int J Mol Sci. 24; 17101. An editorial article discussing the targeting the TGF-β signalling axis in metastatic colorectal cancer. Again, the specific relevance to our current manuscript is unclear to us. 

While we agree that these are important and noteworthy articles, and may arguably be tangentially related to our manuscript, given our model is of colorectal cancer, we would contend that they address topics that are largely outside the scope of our current manuscript. If citing these articles is essential for acceptance, we would respectfully request some further guidance as to what is specifically required and clarification as to why these particular articles should be cited.

---

## [Editor Report · Decision Letter 1]

17 Jul 2024

Development of an in vivo syngeneic mouse transplant model of invasive intestinal adenocarcinoma driven by endogenous expression of Pik3caH1047R and Apc loss

PONE-D-24-19048R1

Dear Dr. Phillips,

We’re pleased to inform you that your manuscript has been judged scientifically suitable for publication and will be formally accepted for publication once it meets all outstanding technical requirements.

Kind regards,

Gianpaolo Papaccio, M.D., Ph.D.

Academic Editor

PLOS ONE

Additional Editor Comments (optional):

The Authors made the minor amendments as requested
---

## [Editor Report · Acceptance letter]

25 Jul 2024

PONE-D-24-19048R1 

PLOS ONE

Dear Dr. Phillips, 

I'm pleased to inform you that your manuscript has been deemed suitable for publication in PLOS ONE. Congratulations! Your manuscript is now being handed over to our production team.

Kind regards, 

on behalf of

Prof. Gianpaolo Papaccio 

Academic Editor

PLOS ONE